# Analysis of English free association network reveals mechanisms of efficient solution of Remote Association Tests

**Olga Valba**[1]*, **Alexander Gorsky**[2,3], **Sergei Nechaev**[4,5], **Mikhail Tamm**[1,6]

**1** Department of Applied Mathematics, MIEM, National Research University Higher School of Economics, Moscow, Russia, **2** Kharkevich Institute for Information Transmission Problems RAS, Moscow, Russia, **3** Moscow Institute of Physics and Technology, Dolgoprudny, Russia, **4** Interdisciplinary Scientific Center Poncelet (IRL 2615, CNRS), Moscow, Russia, **5** P.N. Lebedev Physical Institute RAS, Moscow, Russia, **6** Faculty of Physics, Lomonosov Moscow State University, Moscow, Russia

* ovalba@hse.ru

**Data Availability Statement:** All relevant data are within the manuscript and its Supporting information files.

**Funding:** This work is supported by RFBR grant 18-29-03167. OV is supported within the

## Abstract

We study correlations between the structure and properties of a free association network of the English language, and solutions of psycholinguistic Remote Association Tests (RATs). We show that average hardness of individual RATs is largely determined by relative positions of test words (stimuli and response) on the free association network. We argue that the solution of RATs can be interpreted as a first passage search problem on a network whose vertices are words and links are associations between words. We propose different heuristic search algorithms and demonstrate that in "easily-solving" RATs (those that are solved in 15 seconds by more than 64% subjects) the solution is governed by "strong" network links (i.e. strong associations) directly connecting stimuli and response, and thus the efficient strategy consist in activating such strong links. In turn, the most efficient mechanism of solving medium and hard RATs consists of preferentially following sequence of "moderately weak" associations.

## Introduction

Representation of a large number of interacting agents by a network is one of the most powerful ways of efficient treatment of various types of data in biological, technological, and social systems [1, 2], as well as in cognitive processes. A network is a set of nodes (the elementary indivisible units of a distributed system) and binary relations (links) between them. There is a plenty of ways to build networks in the cognitive science, with various setups relevant for a problem-dependent specific conditions. Historically, semantic networks were used to represent a "knowledge" by establishing directed or undirected semantic relations (graph links) between the "concepts" (graph nodes) [3]. Such networks are useful to study the "intermediate" (or "mesoscopic") scale of organization in the human cognition [4]. However, in attempts to model cognitive processes, it has been realized that the "microscale" network organization, i.e the structure of the detailed concept-to-concept connections, is very important.

framework of the Academic Fund Program at the HSE University in 2020–2021. AG acknowledges the support by the grant 075-015-2020-801 of Ministry of Science.

**Competing interests:** The authors have declared that no competing interests exist.

Advances in graph-theoretic methods of studying cognitive functions are inextricably connected with pioneering works [5–7]. Since then, the number of investigations in the field has grown rapidly (for review, see [8]) and in particular a lot of attention has been paid to the study of large-scale semantic networks. In such networks, words (e.g., nouns) are nodes connected by links indicating semantic relations between them. There is a variety of characteristics of semantic proximity: one can connect the nearest neighboring words in sentences (so-called syntactic networks), one can connect words according to standard linguistic relations between them (synonymy, hyper- or hyponymy, etc.), on their phonetic similarity, etc. Finally, one can assemble networks of words based on various psycholinguistic experimental data.

Large-scale semantic networks possess specific patterns of connectivity, presumably imposed by the growth processes by which these networks are formed. Typically, such networks demonstrate power-law distributions of node degree, such that most network nodes have a low vertex degree, while there are some nodes with a very high degree, playing the role of hubs.

Important and fast-growing area in the field of linguistic networks is related to the so-called "word embedding" [9]. "Word embedding" is a set of language-modeling techniques based on mapping of words to vectors of numbers, usually in a multidimensional Euclidean space. The semantic similarity of two words is defined as a scalar product of the corresponding vectors. Such a procedure results in construction of a complete weighted network of words (each pair of words is connected by a weighted edge, whose weight is the semantic proximity) and the majority of edges have very small weights. Removing all links with weights less than a preset threshold results in a network with nontrivial topological properties. It might be very productive to generalize the "word embedding" ideology to non-Euclidean spaces, in particular to spaces of constant negative curvature which are natural target spaces for scale-free networks [10, 11]. It is known from other applications that such a non-Euclidean embedding might allow to radically decrease the dimensionality of embedding space. To complete this short overview of various theoretic approaches, let us mention that recently several attempts have been made to treat semantic networks as multiplex networks. Such approaches seem to give deeper insight into the formation of mental lexicon [12] and early word acquisition [13].

One particularly interesting class of semantic networks is a network of free associations [14–18]. This class of networks is obtained in the following real experiment. Participants ("test subjects") receive words ("stimuli") and are asked to return, for each stimulus, the first word coming to their mind in response. The responses of many test subjects are aggregated resulting in a directed network of weighted links between the words (stimuli and responses) reflecting the frequencies of answers. The study of these networks has a long history [14, 15]. In what follows we use a free association network constructed in frameworks of the "English Small World of Words" project (SWOW-EN) [16]. The online data collection procedure allowed the authors of [16] to aggregate data for more than 12 000 stimuli words. The data was collected in 2011-2018 and consists of responses of more than 90 000 test subjects. As a result, this network includes many weak, rare associations, which have not been registered in earlier experiments.

We use free associations network to propose some heuristic mechanisms of solving the so-called Remote Associates Tests (RATs). RAT had been invented by S. Mednick in 1962 [22] and was repeatedly used in cognitive neuroscience and psychology [19–21] to study insight, problem solving and creative thinking. In a RAT test subjects are given a set of three stimuli words (e.g. "surprise", "line", "birthday") and are requested to find a fourth "return" word, which is simultaneously associatively related to all three stimuli (in our example it is the word "party"). According to [22], creative ideas arise by forming new combinations of remotely associated elements. These elements are organized in lexical–semantic and associative structures termed "associative hierarchies". Mednick suggested that formation of new links via

remotely associated elements (i.e. creativity), is more pronounced for individuals who have many relatively weak associations. To the contrary, test subjects whose internal lexical-semantic structure consists of fewer and stronger associations (which are typically also common and conventional), are less capable of forming creative ideas. Mednick designates the cases described above as "flat" and "steep" associative hierarchies, respectively. In contrast to this interpretation of creativity, other researchers have suggested that creativity may result not from the difference in the lexical–semantic maps of the subjects but rather form better adaptive control of the thought processes [23–25]. For instance, inhibition and switching have been shown to play an important role in overcoming stereotyping and suppressing prepotent responses in the solution of RATs [26].

Here we use network-analytical techniques to elucidate the process of solving RATs. It has been demonstrated recently [27, 28] that lexical–semantic and associative networks for more creative and less creative individuals actually are substantially different: the networks of more creative individuals are more interconnected, flexible, and robust, thus supporting Mednick's conjecture [22] about the nature of creativity. These observations justify the application of network-based methods for the study of RAT solving, and indeed over the years there were several attempts to approach this problem from the network-analytical perspective. In [29] authors analyzed sequences of guesses, which came to mind during RAT solving. They measured the similarity between guesses, stimuli, and responses using the "Latent Semantic Analysis" [29] and concluded that there are two systematic strategies of solving multiply constrained problems. In the first strategy, generation of guesses is primarily based on just one of three stimuli, while in the second strategy, it is implied that the test subject is making new guesses based partially on his/her previous guesses. In [30] the Metropolis-Hastings search model has been used, which involved the transition probabilities based on geodesic (shortest) distances along the network from the stimuli to the response. The authors underline the importance of association strength between words in the process of RAT solving. The work [31] is devoted to the design, implementation and analysis of a computational solver, which can answer RAT queries in a cognitively inspired manner. In [31] authors developed an artificial cognitive system based on a unified framework of knowledge organization and processing. They took into account associative links between the concepts in the knowledge base and the frequency of their appearance. In the latter work [32], it has been shown that the association strength and the number of associations both have important separated effects on success rate of RAT solving. Finally, the spiking neural network model is proposed in [33]. There, RAT solving is simulated as a superposition of two cognitive processes: the first one generates potential responses, while the second one filters them.

In our study we address two main questions. First, we study the connection between the average hardness of a particular RAT and the position of stimuli and response words on the free association network. We show that the RAT hardness can be predicted reasonably well by examining the network structure. Second, we discuss possible heuristic cognitive search algorithms of solving RATs, and study ways of their optimization.

The paper is organized as follows. We provide a brief characteristic of used datasets: the structural properties of the free association network, and the quantitative definition of RAT hardness. Further we study correlations between the RAT hardness and the relative position of stimuli and response on the free association network. We show that RAT hardness correlates with the aggregated weight of directed bonds stimuli → response, as well as with the aggregated weight of multi-step chains of associations. On the other hand, there are no substantial correlations between RAT hardness and the weights of reverse (response → stimuli) bonds. We argue that such asymmetry means that solving a RAT is a first-passage problem: the correct response is easy to identify as soon as one finds it along a directed path on the network.

Finally, we study various ways of enhancing the probability of a fast solution of a RAT. We argue that search strategies with resetting seem are preferrable to both nearest-neighbor searches, and searches by unlimited chains of associations. Further, we discuss in detail the role of weak associations in the search. In particular, we show that the best strategy for solving easy RATs implies removing all weak associations, and following only the strong ones. In turn, solving medium, and especially hard RATs, in the same way is often impossible. The probability to find a solution of hard RATs gets enhanced when the search runs preferentially along moderately weak bonds (associations). In Discussion we summarize the obtained results and propose possible direction of further investigations.

## Data analysis

We use the free association network described in [16], known as "English Small World of Words project" (SWOW-EN). It is a weighted directed network with $N$ = 12 217 stimuli words. Stimulus materials (cue words) were constructed using a snowball sampling method, allowing authors [16] to include both frequent and less frequent cues at the same time. The final set consists of 12 292 cues (stimuli), the weight of the link indicates fraction of the experiment participants which gave this particular response to a cue (i.e. the conditional probability of a response given a cue). Therefore, the total weight of links going out of each node is less or equal to 1. For our analysis we used the strongly connected component of the SWOW-EN network, the brief summary of the network topological characteristics is given in Table 1.

In Fig 1a we show the distribution of in- and out-degrees of the network. The out-degree distribution (blue) is Poissonian, its average is controlled by the experimental setup: the bigger the number of test subjects per stimulus word, the larger the average degree. In turn, the in-degree distribution $\rho(k)$ (orange), where $k$ is the number of bonds coming into a node, has a

**Table 1. Some topological properties of the SWOW-EN network.**

| | |
|---|---|
| Nodes | 12 217 |
| Mean in/out degree | 31.67 |
| Mean link weight | 0.03 |
| Diameter | 8 |
| Transitivity | 0.08 |
| Percolation threshold | 0.08 |

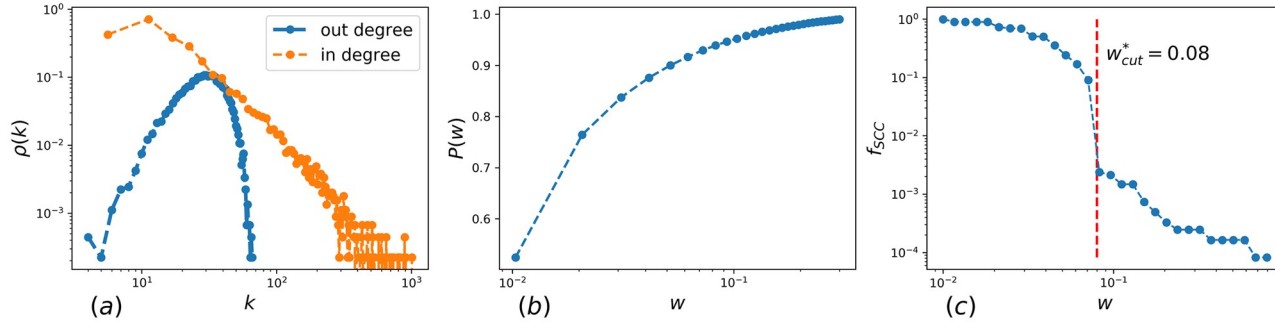

**Fig 1. (a) Distributions of in- (blue) and out- (orange) vertex degrees of SWOW-EN; (b) Cumulative distribution of link weights of SWOW-EN; (c) Fraction of nodes in strongly connected component of SWOW-EN in dependence on the link weight cutoff (see [16]).**

power-law tail, i.e. $\rho(k)\sim k^{-\gamma}$ with $\gamma \approx 3$. Interestingly, such a shape of the in-degree distribution seems to be quite universal: similar values of $\gamma$ are observed for networks in other experiments with English networks [14, 15], and also for the Russian free association networks [17, 18]. In Fig 1b we demonstrate the cumulative distribution $P(w)$ of weights $w$ of links of SWOW-EN (all weights lie within the interval [0.01, 1.00]) and in Fig 1c we depict the size of largest strongly connected component of SWOW-EN as a function of the link weight cutoff, $w_{cut}$. Note that the size of strongly connected component of SWOW-EN collapses at the link cutoff about $w_{cut}^* = 0.08$, which corresponds to removal of 95% of network edges. The SWOW-EN above $w_{cut}^*$ is not percolating anymore, and splits into disjoint components.

In order to characterize the hardness of RATs we use empirical accuracies reported in [34]. Mednick's original versions of the RAT contained 30 items each, and the solution word for each item was sometimes associated with the words in a triad in several different ways. E.M. Bowden and M. Jung-Beeman [34] proposed to use greater number of problems and made the RATs more consistent, that is, the solution word is always related to the specific triad of words. We restrict ourselves to 138 problems (combinations of three stimuli and response) out of 144 studied in [34], for which all four words are present in the strongly connected component of the SWOW-EN network. The hardness of a RAT, for each of 138 problems under consideration, is quantitatively characterized by the fraction $H$ ($0 \leq H \leq 1$) of test subjects who correctly solved it in 15 seconds. Additionally, we divide problems into three broad categories: "easy", "medium", and "hard". The problem is easy if it has been solved in 15 seconds by more than 64% test subjects ($0.64 \leq H \leq 1$), medium if it was solved by 32%÷64% test subjects ($0.32 \leq H \leq 64$), and hard otherwise ($0 \leq H \leq 0.32$). There are 15 easy, 38 medium, and 85 hard problems. For completeness, we provide the hardness of 138 problems used here, and taken from [34] as Supporting information.

## Correlation between average RAT hardness and weights of edges in a free association network

The strength of an association between two given words (vertices) in a free association network, $G$, is described quantitatively by the weight of the corresponding directed link. The whole set of weights is encoded in the weighted adjacency matrix, $W(G)$. The element, $w_{ij}$, of the matrix $W$ is equal to the strength of directed association $i \rightarrow j$ if such association exists, and 0 otherwise (i.e. if the directed link $i \rightarrow j$ is absent).

Our main heuristic assumption is that to solve a RAT problem, a test subject performs a search on a free association network, which is a proxy of a search process happening in memory. Such search process might imply, for example, exploration of all direct associations of all three stimuli words, or following chains of consequently extracted associations starting from stimuli words (such a chain may or may not be limited in length). More sophisticated search strategies can be used as well: for example, one may follow paths on network with preference of weak associations, or one may use some synergy between stimuli words (e.g. choosing words with strong associations with two or more stimuli words), etc. Finally, there exist a possibility that the solution is found but it is not recognized as the right one.

In order to test the basic hypothesis that the RAT solution is governed by some search process on the free association network, we study correlations between the RAT hardness and probabilities of finding a solution in various simple search strategies.

We begin with a simplest possible one-step strategy: (i) choose one of the stimuli words at random, (ii) jump to its neighbor along the directed link on the free association network (the jump probability is given by the link weight), (iii) check whether the solution is correct. The

probability of finding a correct answer in such a strategy is, obviously,

$$p_0(\alpha) = \frac{1}{3}\left(w_{s_\alpha^1, r_\alpha} + w_{s_\alpha^2, r_\alpha} + w_{s_\alpha^3, r_\alpha}\right) \qquad (1)$$

where $\alpha$ enumerates different RAT problems ($1 \le \alpha \le 138$), the indices $s_\alpha^1, s_\alpha^2, s_\alpha^3$ designate three stimuli words (vertices of the network) for a given problem $\alpha$, and the correct response is a network vertex with the index $r_\alpha$. Thus, $w_{s_\alpha^j, r_\alpha}$ is the weight of the directed bond $s_\alpha^j \to r_\alpha$, where $j = 1, 2, 3$.

Another simple hypothetic model is as follows. Consider a search on the network as a sequence (Markov chain) of associations: one generates a random walk trajectory with jumping probabilities equal to $w_{ij}$, starting from one of stimuli words. In that case, the probability $\pi_{s,r}$ of reaching the response word from one of the stimuli is

$$\pi_{s,r} = w_{s,r} + \sum_{k \ne r} w_{s,k} w_{k,r} + \sum_{k,l \ne r} w_{s,k} w_{k,l} w_{l,r} + \cdots = \left[\frac{W^-}{I - W^-}\right]_{s,r}, \qquad (2)$$

where $W^-$ is the adjacency matrix $W$, in which all weights $w_{r,i}$ out of the resposne word $r$ are set to zero, which guarantees that only first passage of the response word is counted, while all the other elemenrts of matrix $W$ are preserved.

If the starting stimulus word is chosen at random, the resulting probability of solving the task by the proposed mechanism reads:

$$p_1(\alpha) = \frac{1}{3}\left(\pi_{s_\alpha^1, r_\alpha} + \pi_{s_\alpha^2, r_\alpha} + \pi_{s_\alpha^3, r_\alpha}\right) \qquad (3)$$

Every search is restricted in time. Therefore, the Markov chain representing the search on the network, should be finite. Thus, it seems reasonable to truncate the maximal length of search trajectories: if the search is not completed during the allowed time interval, we stop the search and start the new one from the same stimuli. Such a strategy resembles the random search with resetting [35, 36]. In case of a random resetting, the probability of solution in one search, given a stimulus $s$ and a response $r$, can be written as follows

$$\pi_{s,r}(\lambda) = \left[\frac{W^-}{I - \lambda W^-}\right]_{s,r}, \qquad p_\lambda(\alpha) = \frac{1}{3}\sum_{i=1..3} \pi_{s_\alpha^i, r_\alpha}(\lambda) \qquad (4)$$

where $\lambda$ is an additional weight (probability of keeping the search as opposed to stopping), associated with each step. As a result, the probability of each $N$-step search is multiplied by an additional factor $\lambda^{N-1}$, and, for any $\lambda < 1$, the long searches are suppressed.

In Fig 2a–2c scatter plots providing correlations between various search strategies and the empirical hardness, $H$, are shown. In particular, Fig 2a presents the correlation between the average association weight (4) from stimuli words to the response, $p_{\lambda=0}$, and $H$; Fig 2b—the correlations between the estimated probability of the random walk with resetting, $p_{\lambda=1/2}$ and $H$; (c)—the same as (b) for $p_{\lambda=1}$ and $H$. Dashed lines show slopes of the linear regression analysis, the corresponding Pearson correlation coefficient is designated by $\rho$. In all cases we observe sufficiently large values of $\rho$, which confirms our hypothesis that the RAT hardness correlates with relative locations of words in the associative network.

There is, meanwhile, another interesting question. The simplest strategies suggested above imply that solving RATs is a first-passage problem. This implies that when the solver finds a solution, he immediately recognizes it. Is it indeed the case? We have not been able to check this assumption directly but there is an indirect argument in support of it: if the task of

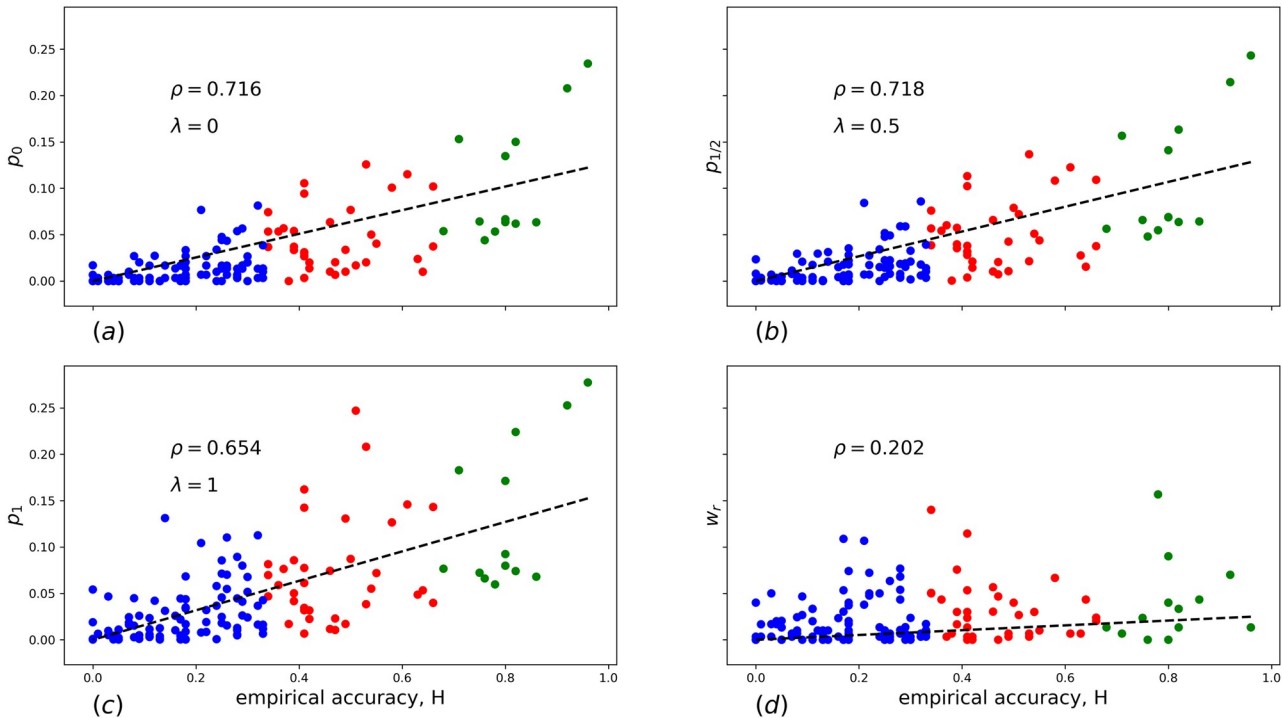

**Fig 2.** The scatter plots of the empirical accuracy of the RAT solution versus following variables: (a) the average association weight from the stimuli words to the response $p_0(\alpha)$; (b) the estimated probability of random walk with resetting with $\lambda = 1/2$, $p_{1/2}(\alpha)$; (c) the estimated probability of unlimited ($\lambda = 1$) random walk, $p_1(\alpha)$; (d) the average association weight from the response to the stimuli words $w_\alpha$. In all figures $\rho$ is the Pearson correlation coefficient.

recognizing of an already found response word was hard, it should have lead to consequences which we do not see in the data.

The argument goes as follows. If people make mistakes recognizing an already found response, they should do it differently for different RATs. In some cases an already found response should be easy to recognize, in some others it should be more difficult. How to predict, for which RATs correctly recognizing the solution is hard? It seems self-evident, that the inverse association weights $w_{r_\alpha, s_\alpha^i}$ should play a role in such prediction. That is to say, if the association from response word to stimuli is strong, the response word is easily recognized as correct, while if that inverse association is weak, there is a larger probability that a test subject finds the response word but fails to recognize it as a correct answer.

Therefore, we expect that if the task of recognizing the correct solution contributes meaningfully to the overall probability of solution then there should be correlation between the hardness of a RAT and the inverse weight $w_{r_\alpha, s_\alpha^i}$. In Fig 2d we show a scatter plot of the average inverse weight, $w_\alpha = \frac{1}{3}\sum_{i=1..3} w_{r_\alpha, s_\alpha^i}$ versus the RAT hardness. Clearly, the relation is very weak, much weaker then the relation with direct weights. In our opinion, this indicates, although indirectly, that the problem of recognizing an already found solution is of secondary importance, as compared to the problem of finding the solution, and one can indeed treat solution of a RAT as a first passage problem.

Although Fig 2 provides much important information, it does not reveal which search strategy is preferential. Indeed, $p_\lambda$ gives only a probability of finding a solution by a single Markov chain search, regardless its length. In reality, since the search is limited in time, the test subject might have enough time to try: either ten 1-step searches, or only one 10-step search.

Moreover, one expects that there is a high variability in how different test subjects treat the RAT problems. Thus, the question "How to maximize the probability of solving a RAT?" seems to be more reasonable than the question "How do people solve RATs on average?"

## Enhancing the probability of correct solution

Here we discuss strategies which maximize the probability of solving a RAT problem. In particular, we are interested whether the heuristic optimal strategy depends on a RAT hardness. Clearly, two simplest strategies, (1) and (4), outlined above, have significant drawbacks from that point of view. Searching only in the immediate proximity of a stimuli might be sufficient to solve easy RATs but for hard RATs there typically is no direct associations (direct links) between stimuli and response, thus solving a problem by such a strategy is simply impossible. In turn, searching via a random walk on a network might lead to excessively long solution times.

Therefore, it seems natural to construct a search algorithm on a SWOW-EN network in a way that search trajectories, while not artificially constrained to nearest neighboring nodes of the stimuli, still are not fully random walks. One can think of these trajectories as of random walks in an external attractive potential, which guarantees that the test subject does not ever loose the stimuli words from his/her mental view. Such a strategy seems to be in agreement with the experimental data on sequences of guesses provided in [29] and discussed briefly in the Introduction.

### Search with an attraction to the stimuli

The proposed heuristic search algorithm on SWOW-EN is organized as follows. At time $t = 0$ there are three stimuli (nodes of the network) $s^i$, $i = 1, 2, 3$ which are considered active. At the next time step, $t = 1$, one of nearest neighbors of the active nodes, $x$, is activated with probability $P(x)$ proportional to the sum of links from active network nodes towards it, i.e.,

$$P(x) = \frac{\sum_a w_{a,x}}{\sum_k \sum_a w_{a,k}}, \quad (x, k) \in \mathbb{NN}(\{a\}), \tag{5}$$

where index $a$ enumerates active nodes, while index $k$ enumerates all possible target nodes from the set of nearest neighbors of the active ones $\mathbb{NN}(\{a\})$.

Thus, at time $t = 1$ there are four activated words. If the newly activated word is the correct response, $r$, the search is completed. If it is not, on the next step, $t = 2$, we activate a new neighboring word with the probability given by (5) but for the fact that now there are 4 instead of 3 active words in the set $\{a\}$. Simultaneously, we deactivate the word which was activated on a previous step, and mark it as checked, so that it will not be ever activated again. Now we check if the newly activated word is the correct response, $r$. If yes, we exit the search, if not, proceed recursively as described above. At all times except $t = 0$ there are exactly 4 active words, and by time $t$ exactly $t$ different possible response words have been checked.

By such rules we mimic a search strategy according to which the activated word, if it is not a response (i.e. a correct answer), still can affect the search trajectory leading to the answer. The fact that three stimuli remain always active at each time step, while intermediate guess words are activated and deactivated during a search, guarantees that there is an effective "attraction" of the search trajectory to the set of stimuli, which can be interpreted as a permanent "memory" about the initial stimuli.

The search algorithm stops if either the correct answer is found or if the number of search attempts exceeds $t_{max}$. We performed $10^4$ runs of this algorithm for each RAT, and counted the fraction of runs leading to a correct response. This fraction, which we call *model accuracy*,

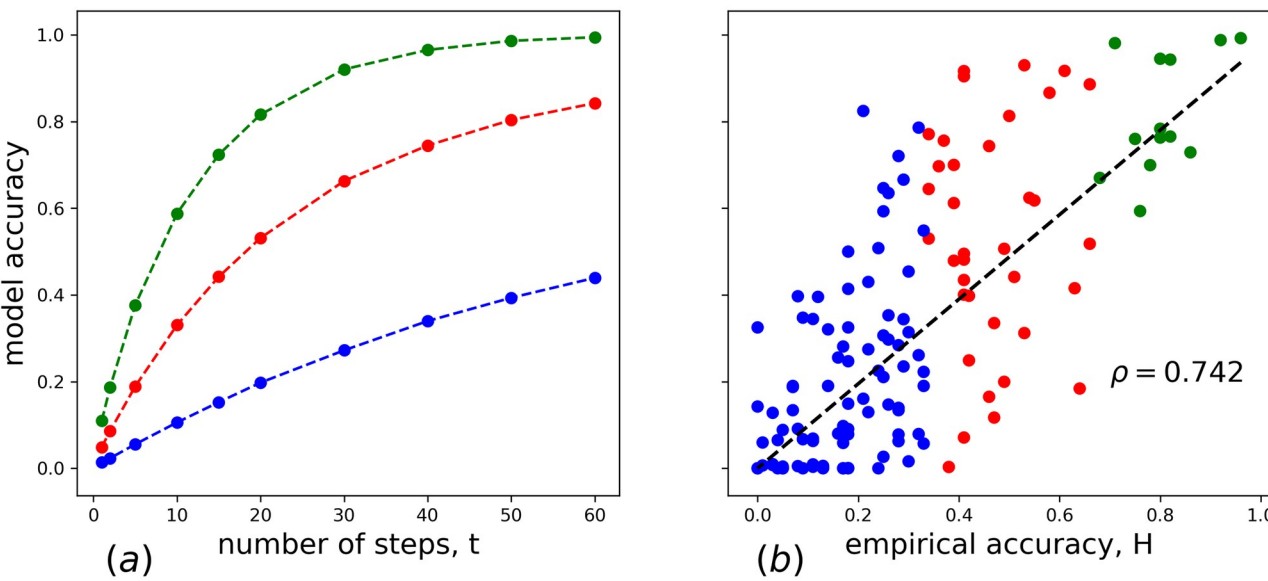

**Fig 3.** (a) The dependence of model accuracy on the number of search attempts (steps on SWOW-EN); (b) the scatter plot of model accuracy versus empirical accuracy of the RAT. The model accuracy is averaged over $10^4$ simulations. The dashed line in (b) is the best linear fit to the data, $\rho$ is the corresponding Pearson correlation coefficient.

is a natural measure of the performance of the algorithm. Naturally, it is a monotonously increasing function of $t_{\max}$ (see Fig 3a) and at $t_{\max} = 20$ the average accuracy of hard, medium and easy RATs numerically coincides with corresponding typical probabilities of correctly solved RATs in 15 seconds [34].

In Fig 3b the scatter plot of the empirical RAT hardness versus the model accuracy is shown. The strategy presented in this section has larger correlation between empirical and model accuracy ($\rho = 0.742$) than the more naïve strategies discussed in the previous section (maximal $\rho = 0.718$). Besides, the new algorithm also has a benefit of working in finite time, which makes more sense from the psychological point of view, being also compatible with results known about the RAT solving mechanism from psychological sources [29]. We believe therefore that this heuristic strategy can work as a reasonable first proxy for the real mechanism of RAT solving. However, some simple generalizations of this strategy lead to further enhancement of the solvability of RATs, especially medium and hard ones. We discuss these generalizations in the subsequent sections.

### Activation algorithm with a threshold

Consider now a modification of our heuristic search algorithm described above. It is known that many activation processes need a certain threshold (minimal activation impulse) to get triggered. In the psycholinguistic context, the importance of the association strength and the number of associations in the search processes is well known [32]. In the spirit of the work [32] we introduce an activation threshold to our model: we modify (5) as follows

$$
P(x) = \begin{cases} \dfrac{\sum_a w_{a,x}}{\sum_k \sum_a w_{a,k}} & \text{if } \sum_a w_{a,x} \geq \tau, \quad (x,k) \in \mathbb{NN}(\{a\}) \\ 0 & \text{otherwise} \end{cases}, \tag{6}
$$

i.e., demand that target $x$ can be activated only if the sum of activation signals exceeds a certain threshold, $\tau$.

Consider now how the solvability of a RAT should depend on the value of threshold $\tau$. First of all, in any experiment, including SWOW-EN, the relative accuracy of measuring very weak links is low. Indeed, if an association is mentioned by just 1 or 2 test subjects, which corresponds to $w \approx 0.01$ in terms of network weights in SWOW-EN network, the relative error of meaurement is of order one. Inevitably, significant part of such weak associations are just experimental noise. Increasing the threshold up from 0 effectively suppresses these very weak and often erroneous associations. As a result, one expects that initially, for small enough $\tau$ the solvability of RATs should grow with growing $\tau$.

The behavior for larger $\tau$ is less easy to predict since it depends on a structure of a particular RAT. If the solution of RAT depends on finding a very strong association between the stimulus and the response (or the sequence of such strong associations), then increasing $\tau$ should be beneficial: removing weak links, we make strong ones even more essential. In turn, if the solution of RAT is reached by following the chain of moderately weak bonds one expects the probability of solution to go through a maximum. For small $\tau$ the solvability of RATs increases with increasing $\tau$ due to removal of the noisy weak links. But with further increasing $\tau$ the solvability drops down when essentially important link (or links) cannot be activated due to a high value of threshold.

In Fig 4a, 4c and 4e we show average predicted accuracy for different hardness categories as a function of $\tau$ ($\tau \in [0, 0.1]$), while in Fig 4b, 4d and 4f we show corresponding average times needed to solve easy/medium/hard RATs. For each $\tau$, the accuracy is averaged over $10^4$ simulations of all RATs in a given hardness category (easy/medium/hard).

Solvability of easy RATs (Fig 4a) grows monotonously and approaches unity with increasing $\tau$. Indeed, in easy RATs there is at least one strong directed link from a stimulus to the response. The situation is different for medium and hard RATs. In this case the accuracy goes through a maximum at around $\tau_m = 0.04$ (which is still below the percolation threshold corresponding to $w^* = 0.08$—see Fig 1c). The probability of a correct solution at the maximum, $P(\tau_m)$, significantly exceeds the result of both a no-threshold model and of a model where only strong links are retained. Compared to the last model (with only strong links left), the gain is by a factor of 1.3 for medium RATs and by a factor of almost 2 for hard RATs. This means that moderately weak links are instrumental for solving medium and hard RATs: eliminating these moderate links decreases the solvability, and, as shown in Fig 4b, 4d and 4f, increases the mean length of search trajectories.

## Enhancing the role of weak associations

The result of previous section gives rise to the following natural question. Is it possible to enhance the solvability of medium/hard RATs further by preferentially following moderately weak links? To check this, let us, apart from removing weak links, remove also the strong ones. That is to say, introduce a new adjacency matrix $\bar{W}$ with matrix elements

$$\bar{w}_{ij} = w_{ij} H(1 - w_{max}), \tag{7}$$

where $H(x)$ is the Heaviside function, and $w_{max}$ is the upper cutoff parameter. In Fig 4c and 4e we show the model accuracy and mean length of solving trajectories for hard and medium RATs for the null model (6) with no upper cutoff ($w_{max} = 1$) and for the same model with $w_{max} = 0.05$. We see that by introducing a cutoff we can significantly increase the maximal accuracy for hard and medium RATs, roughly by factors 1.3 and 1.1, respectively. Suppressing very strong associations is beneficial for the solution of medium and, especially, hard RATs.

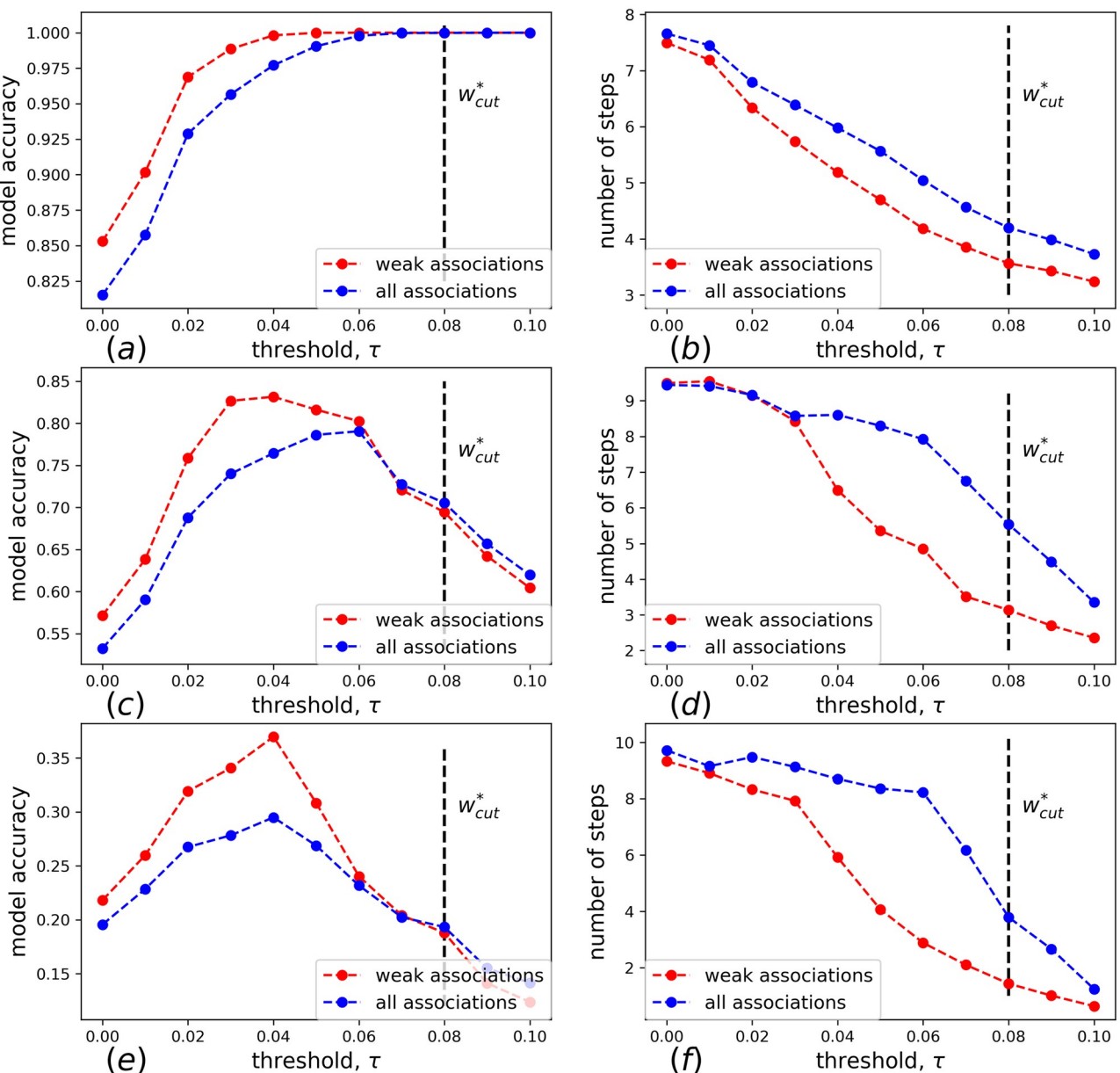

**Fig 4.** (a) The dependence of the model accuracy on the strength of the threshold $\tau$ for the RAT of different categories. The accuracy is calculated over $10^4$ simulations and averaged over all RATs in a given hardness category. (b) The dependence of the mean length of solving trajectories on the threshold strength.

Interestingly, despite the fact that introducing a cutoff removes some associations from the network, the mean length of solving trajectories decreases in the model with cutoff as compared to the null model. This indicates some sort of path length optimization in the cutoff model. Indeed, paths from stimulus to response in the null model often consist of several intermediate activated words, connected by strong associations. In the cutoff model typical solutions trajectories go along shorter sequences of weaker bonds. Apparently, judging by the model accuracy, the second strategy results in a better success rate. These results once again indicate the crucial role of moderately weak associations in the solution of medium and hard RATs.

## Discussion

In this paper, we apply the network approach for studying the psycholinguistic mechanisms of solving Remote Association Tests (RATs). Our treatment is based on available open data on the network of free associations in English language (SWOW-EN) [16], and on standardized concept of hardness of RATs [34].

We use network of free associations as a proxy of the mental lexicon structure. Clearly, the real organization of the language is much more complex: one may think of it as a multiplex network with layers corresponding to grammatical connections, phonetic relations, synonimity, co-occurence in texts, etc. We checked several semantic networks (see S2 Fig) and shown that search on free associations network is much more predictive for the experimentally observed hardness of RATs than search on other linguistic networks. The reason for that, in our opinion, is that association network is, to some extent, different in nature from other linguistic networks. While constructing personal association networks people effectively integrate information from all other networks of the language: phonetical, grammatical, corpus-based, etc., into a single internal map. As a result, although association network is, of course, less informative than the whole multiplex of linguistic networks, it seems, at least from the psycholinguistical point of view, to be more informative than any other separate (one-layer) linguistic network.

We quantitatively characterize the correlation between the hardness of a particular RAT and the location of stimuli on the directed network of free associations. Hardness of a RAT turns out to be strongly correlated with the aggregated weight of bonds from the stimuli to the response, as well as with the aggregated weight of multistep chain of consecutive associations. On the other hand, there is no significant correlation between the RAT hardness and weights of reverse (response → stimuli) bonds.

We investigate the efficacy of RAT solution using an activation algorithm which resembles the random walk in a potential well with attraction to the stimuli words of the RAT. We show that while for easy RATs the solution is mostly governed by strong associative bonds from stimuli to response, the solution of medium and especially hard RATs, is mostly determined by moderately weak bonds, i.e. bonds with weights about $w = 0.04 \pm 0.01$. Indeed, for the threshold model, while neglecting very weak bonds is beneficial for the solution efficacy, neglecting moderately weak bonds suppresses the efficacy of finding the correct response. Even more, one can further enhance the solution probability for medium and hard RATs by removing strong bonds with weights larger than $w = 0.05$.

Thus, "very weak" and "moderately weak" bonds behave differently in our consideration. That could be related to inevitable errors in measurement of very weak bonds in free association network experiments, so that significant number of registered very weak bonds are just experimental noise. We expect that the efficacy of the solution might be additionally increased by replacing the experimental free association network with a "cleaned-up" one in the spirit of [37]. From a more general perspective, the importance of weak associations in the solution of RATs seems to be an example of the ubiquitous importance of weak ties in social sciences [38].

This gives rise to a very interesting question related to the long-standing polemic in the theory of creativity. It is clear from our results that enhancing weak associations is beneficial for the solution of RATs, and, more generally, for human creativity. However, it is not clear how exactly do creative people enhance such weak associations. One option, suggested in [22] and supported by recent work [27, 39] is that mental lexicon of creative people is intrinsically different from that of less creative ones in a sense that the weight distributions of their individual association networks are more flat, i.e., strong associations in their individual networks typically have smaller weights than on average in the population, and weaker associations have

relatively higher weights. Another option, suggested in particular in [23–25] is that it is not the difference in individual association networks which is important, but rather a difference in search algorithms that people use: that creative people are capable of suppressing strong and stereotypical associations at will.

In this paper we study only average aggregated data describing association networks and hardness of the RATs. As a result, our main finding, "moderately weak bonds are essential for solving medium and hard RATs", is compatible with both interpretations stated above. It would be extremely interesting to distinguishing between these two conjectures, but that needs a much more fine-grained data on individual test subjects. In our opinion, it might be a very interesting topic for further experimental and analytical work.

## Supporting information

**S1 Fig. Complementary cumulative distribution functions of in degree distribution and fitted power law, lognormal and truncated power law distributions.** We compared the fits of three candidate distribution using Python package powerlaw [40], results of likelihood-ratio test are presented in S1 Fig.
(TIF)

**S2 Fig. The scatter plots of the empirical accuracy of the RAT problems versus the average semantic similarity from the stimuli words to the response.** We used pre-trained vector representations for words from different model: (a) word2vec GoogleNewsvectors [9]; (b) Fast Text Wiki News [41]; (c) Glove Wikipedia + Gigaword [42]; (d) Glove Twitter [42]. In all figures $\rho$ is the Pearson correlation coefficient.
(TIF)

**S1 Table. Log likelihood ratios for different compared distributions.** The best fit to truncated power law $p(k) = k^{-\alpha} e^{-\beta k}$ is established with the respective parameters: $\alpha = 2.3$, $\beta = 0.002$.
(PDF)

**S2 Table. Hardness data for Remote Association Tests.** The list of 138 RATs we have used in our research and their hardness according to the data of [34].
(PDF)

**S1 Text.**
(PDF)

## Acknowledgments

The authors are grateful to O. Morozova, A. Poddiakov and D. Ushakov for introducing us to the field of psycholinguistics, and to V. Avetisov, I. Kasyanov, E. Patrusheva, N. Pospelov, S. Tolostoukhova, and M. Zvereva for many interesting discussions on the structure and properties of various networks of free associations.

## Author Contributions

**Conceptualization:** Olga Valba, Alexander Gorsky, Sergei Nechaev, Mikhail Tamm.

**Investigation:** Olga Valba, Alexander Gorsky, Sergei Nechaev, Mikhail Tamm.

**Methodology:** Olga Valba, Sergei Nechaev, Mikhail Tamm.

**Validation:** Olga Valba.

**Writing – original draft:** Olga Valba, Mikhail Tamm.

**Writing – review & editing:** Olga Valba, Alexander Gorsky, Sergei Nechaev, Mikhail Tamm.

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
