## [Decision Letter · Decision Letter 0]

5 Jan 2021

PONE-D-20-34630

Analysis of English free association network reveals mechanisms of efficient solution of Remote Association Tests

PLOS ONE

Dear Dr. Valba,

Thank you for submitting your manuscript to PLOS ONE. After careful consideration, we feel that it has merit but does not fully meet PLOS ONE’s publication criteria as it currently stands. Therefore, we invite you to submit a revised version of the manuscript that addresses the points raised during the review process.

The reviewers have assessed the manuscript positively, however they do raise some concerns and offer some suggestions for improvement. Please carefully address all the comments by the reviewers.

We look forward to receiving your revised manuscript.

Kind regards,

Leto Peel

Academic Editor

PLOS ONE

Journal Requirements:

2.We suggest you thoroughly copyedit your manuscript for language usage, spelling, and grammar. If you do not know anyone who can help you do this, you may wish to consider employing a professional scientific editing service.  

Reviewers' comments:

Reviewer's Responses to Questions

**Comments to the Author**

1. Is the manuscript technically sound, and do the data support the conclusions?

Reviewer #1: Yes

Reviewer #2: Yes

Reviewer #3: Yes

2. Has the statistical analysis been performed appropriately and rigorously? 

Reviewer #1: Yes

Reviewer #2: Yes

Reviewer #3: Yes

3. Have the authors made all data underlying the findings in their manuscript fully available?

Reviewer #1: No

Reviewer #2: Yes

Reviewer #3: Yes

4. Is the manuscript presented in an intelligible fashion and written in standard English?

Reviewer #1: No

Reviewer #2: Yes

Reviewer #3: Yes

5. Review Comments to the Author

Reviewer #1: PONE-D-20-34630

Analysis of English free association network reveals mechanisms of efficient solution of Remote Association Tests

Summary:

The authors report a series of computational simulations on the SWOW free association network to investigate search strategies involved in the solving of Remote Associates Task. They find that the difficulty of the RAT problem is correlated with structure of the test/solution words in the SWOW network, and examined how the mechanisms of solving RATs could be re-interpreted as search algorithms implemented on a network structure.

Overall comments:

I am very supportive of this work as it is a neat demonstration of how psychology can benefit from leveraging both the mathematics of networks and the availability of large-scale behavioral data. The mathematical simulations appear to be properly done; however, I will leave this for reviewers with more specific expertise to comment on the implementation. My comments will focus more on conceptual issues that the authors should address before the manuscript can be accepted.

1. More detail about the SWOW network (pp. 3-4). Because the simulations of search strategies are conducted on this network, it is important to provide more details about the structure of this network and the properties of the dataset used to construct the network. For instance, do the 12k nodes include the responses to the cue words used in the free association task? How were the weights between pairs of nodes determined (e.g., what does w = 1.0 represent in this context)? Table 1 could also include mean in-degree and out-degree.

2. As reviewed in the introduction, there are many possible search strategies that could be implemented to solve the RATs. The authors seemed to have selected a few that made intuitive sense to them to test. My question to the authors is to what extent are these search strategies psychologically plausible given what is known in the psychological literature about problem solving?

3. I did not quite understand how the lack of a correlation between inverse association weight and difficulty of the RAT led to the argument that the problem of solution recognition is not relevant in this situation (p. 6) and how the simulations with the activation threshold led to the conceptual claim that associative strengths of the links modulated the solvability of the RAT (p. 8). It may be simply be that clearer explanations are needed in these sections.

4. Could the authors provide more discussion theoretical and conceptual implications of their findings? For instance, the RAT is commonly used to evaluate creativity levels among people. Do their findings suggest that creativity can be attributed to the use of appropriate search strategies given the hardness of the RAT (a process-based explanation) or due to structural differences in one’s mental lexicon structure (as proposed by Kenett et al.)?

5. Finally, may I suggest the following references to be included in the paper; justification is provided in parentheses.

Siew, C. S. Q., Wulff, D. U., Beckage, N., & Kenett, Y. (2019). Cognitive Network Science: A review of research on cognition through the lens of network representations, processes, and dynamics. Complexity, 1-24.

[this review is viewed as the state of the art in summarizing the literature on cognitive network science]

Monaghan, P., Ormerod, T., & Sio, U. N. (2014). Interactive activation networks for modelling problem solving. In Computational models of cognitive processes: Proceedings of the 13th Neural Computation and Psychology Workshop (pp. 185-195).

[the authors appear to have conducted a similar set of simulations with respect to the RAT; chapter also contains relevant references about the existing literature on problem solving and creativity]

Reviewer #2: The authors produced an interesting investigation of the mental processes exploring linguistic knowledge in the human mind in order to solve the remote association task (RAT), e.g. a game where a person is given three cues and needs to find a target concept related to all these cues. The authors operationalise mental search through simple Markov chain random walks and more complex activation spreading models over a weighted network of free associations. The authors find activation spreading provides better results than MC walks in efficiently solving the RAT problem, providing additional quantitative evidence for the relevance of activation spreading modelling in lexical processing. The authors also find that weaker free associations are important for solving harder, rather than easier, RAT problems, in line with previous theories about fluid intelligence and creativity.

The manuscript is well written, clearly motivated and nicely outlined. The authors have to be praised for their work and transdisciplinary efforts in providing several modelling approaches to cognitive data. I only have a few minor suggestions that might slightly improve some passages. The manuscript would nicely suit the multidisciplinary audience of PLOS ONE interested in networks and cognitive data.

For the above reasons, I recommend publication of this manuscript on PLOS ONE after (very) minor revisions.

--- Comments:

Line 120 – How did the authors perform the statistical fitting of the degree distribution with a power-law? The authors should report goodness-of-fit values and argument why the power-law distribution is the best fitting model in comparison to other alternatives for heavy-tailed distribution. Notice that this is easy to fix with packages like powerRlaw in R or powerlaw in Python. Notice also that usually the cumulative distribution shows less noise on the tail, making the fitting more robust.

Line 166 – I am not sure but is the one in equation 1 really a probability? The strengths are normalised between 0 and 1 or are they simply counting the amount of people who performed that associations (as I guess it is custom in the SWOW dataset?). A minor rewording might help here.

Line 268 – Please define the “model accuracy”.

Discussion – In the Discussion the authors should address a limitation of their approach: that search in semantic memory is driven only by free associations and not by other types of linguistic relationships, e.g. feature sharing, synonyms, etc.. This assumption should be stated but also mitigated by the fact that free associations already possess a “multilayer” nature, as they can be considered not only as coming from memory patterns but also from the other approaches. But would we expect similar results if we used another type of semantic cognitive network?

Discussion – Does the search with an attraction implement any optimisation of shortest path length/strength? This would nicely relate the search strategy implemented by activation spread to a process of network economy, where mental paths should not “travel around that much” on the network as an indirect consequence of activation spreading across neighbours.

Typos:

Multiplexes -> multiplex networks (notice multilayer networks are more general models than multiplex networks)

An unified approach –> a unified approach

Very week -> very weak

Reviewer #3: The authors present an analysis of the Remote Associations Test (RAT). Basically, they describe the variance in RAT items (why certain items are more difficult to solve) using the network/graph of word associations that is based on a large-scale survey.

Overall, I found their analyses rigorous. The authors use different approaches to explain why certain test items are solved more easily. My main issue with the paper is that it is basically comprised of very sophisticated correlation studies. This is of course not a bad thing by itself, but it does mean that the conclusions one can draw from the results about the causal links between what makes RAT items easy to solve are rather limited. It is my impression that the authors do not communicate this fundamental limitation well enough, which could lead to false interpretation of their findings.

The analyses mix between two highly similar questions: 1) Can the solution word be reached from the stimuli words on the graph?; and 2) Is the location of the solution word on the graph relative to the stimuli words explain why certain stimuli words are more difficult to solve than others?

The many good analyses provided in this paper actually answer the first question, but claim (rather implicitly) to answer the second question. I admit that this is a fine distinction, but I believe that the authors themselves are aware of it (L196-L209). In that sense, the first analysis which computes the aggregated weights as a correlate of RAT "hardness" is more credible than the later analyses of activation threshold. It is clear in these later analyses that "any" word could be reached (a sort of over-fitting of the model the authors present if you would like), but the question of how do human participants recognize the solution to be true is unanswered, and perhaps cannot be answered using these models.

I believe the paper would benefit greatly from openly pointing to its own fundamental limitations, and prefer cautious conclusions rather than strong ones (e.g., remove L269-L270, but also elsewhere). In addition, the analyses could have been made much stronger if there was any control or Null model to compare the results to. For example, if the authors could use a graph/network which is not based on associations (e.g., from word embedding nearest neighbors, or words-co-occurrence), and show that these graphs have far smaller correlations that the association graphs, it would support the superiority of the information encoded in human associations.

In general I would say that reporting .7 correlation of the most basic model is "too high to believe". This remark is not meant to doubt the integrity of the authors of course. I simply find it hard to believe that a hallmark of human cognition, RAT that is supposed to capture the ability of problem solving and creativity, can be predicted to a large extent by aggregated statistics of word associations. This is after all a psychological variable, and it is very rare in psychological research to reach such correlations with a single predictor. I believe that if the authors could think of this remarkable result more critically, they would be able to convey their results and conclusions more cautiously and accurately.

Lastly, I find the survey the authors write in L36-L44 interesting, but completely irrelevant for the rest of the paper. In its current location in the manuscript it unnecessarily deviates the reader's attention. I would ask the authors to remove it, change its position, or better explain why it is necessary in the context of this paper. Same goes for the last two paragraphs of the discussion. Although I understand the wish of the authors to promote their own future work (which sounds very interesting), there is no actual link between the current work and their suggested spectral analysis. I would like the authors to either remove these paragraphs, or do a better job in explaining why the discussion of spectral analysis is relevant in the context of the current work (in my opinion it is not).

Typos and Misc.

- The actual RAT items are based on Bowden and Jung-Beeman set, which differs slightly from the original RAT. The authors would do right if they will make this point clearer in the text.

- L200: the "i" in "invert".

- L223: by > be

- L305: the "p" in "preferentially"

- L346: paople > people

- Figure 2: the "hardness" axis is in fact "easyness". The larger values represent easier items.

- Figure 3b: I believe the x-axis should be the same as in Figure 2, and not "accuracy".

6. PLOS authors have the option to publish the peer review history of their article (what does this mean?). If published, this will include your full peer review and any attached files.

Reviewer #1: No

Reviewer #2: No

Reviewer #3: **Yes: **Haim Dubossarsky

---

## [Author Response · Author response to Decision Letter 0]

19 Feb 2021

Please find the response in attachment

---

## [Decision Letter · Decision Letter 1]

10 Mar 2021

Analysis of English free association network reveals mechanisms of efficient solution of Remote Association Tests

PONE-D-20-34630R1

Dear Dr. Valba,

We’re pleased to inform you that your manuscript has been judged scientifically suitable for publication and will be formally accepted for publication once it meets all outstanding technical requirements.

Kind regards,

Leto Peel

Academic Editor

PLOS ONE

Additional Editor Comments (optional):

Reviewers' comments:

Reviewer's Responses to Questions

**Comments to the Author**

1. If the authors have adequately addressed your comments raised in a previous round of review and you feel that this manuscript is now acceptable for publication, you may indicate that here to bypass the “Comments to the Author” section, enter your conflict of interest statement in the “Confidential to Editor” section, and submit your "Accept" recommendation.

Reviewer #1: All comments have been addressed

Reviewer #2: All comments have been addressed

2. Is the manuscript technically sound, and do the data support the conclusions?

Reviewer #1: Yes

Reviewer #2: Yes

3. Has the statistical analysis been performed appropriately and rigorously? 

Reviewer #1: Yes

Reviewer #2: Yes

4. Have the authors made all data underlying the findings in their manuscript fully available?

Reviewer #1: No

Reviewer #2: Yes

5. Is the manuscript presented in an intelligible fashion and written in standard English?

Reviewer #1: Yes

Reviewer #2: Yes

6. Review Comments to the Author

Reviewer #1: The authors have done a good job of addressing the concerns of this reviewer. Looking forward to seeing this in press.

Reviewer #2: The authors took great care in addressing my comments and the feedback provided by other reviewers. Considering the revised manuscript and the new material included as Supporting Information, I consider this manuscript as being quite relevant for the readership of Plos One interested in computational linguistics and complex networks.

7. PLOS authors have the option to publish the peer review history of their article (what does this mean?). If published, this will include your full peer review and any attached files.

Reviewer #1: No

Reviewer #2: **Yes: **Massimo Stella

---

## [Editor Report · Acceptance letter]

25 Mar 2021

PONE-D-20-34630R1 

Analysis of English free association network reveals mechanisms of efficient solution of Remote Association Tests 

Dear Dr. Valba:

I'm pleased to inform you that your manuscript has been deemed suitable for publication in PLOS ONE. Congratulations! Your manuscript is now with our production department. 

Kind regards, 

on behalf of

Dr. Leto Peel 

Academic Editor

PLOS ONE